# Enantioselective [3+3] atroposelective annulation catalyzed by *N*-heterocyclic carbenes

Changgui Zhao[1], Donghui Guo[1], Kristin Munkerup [2], Kuo-Wei Huang [2], Fangyi Li[1] & Jian Wang [1]

Axially chiral molecules are among the most valuable substrates in organic synthesis. They are typically used as chiral ligands or catalysts in asymmetric reactions. Recent progress for the construction of these chiral molecules is mainly focused on the transition-metal-catalyzed transformations. Here, we report the enantioselective NHC-catalyzed (NHC: *N*-heterocyclic carbenes) atroposelective annulation of cyclic 1,3-diones with ynals. In the presence of NHC precatalyst, base, Lewis acid and oxidant, a catalytic C–C bond formation occurs, providing axially chiral α-pyrone−aryls in moderate to good yields and with high enantioselectivities. Control experiments indicated that alkynyl acyl azoliums, acting as active intermediates, are employed to atroposelectively assemble chiral biaryls and such a methodology may be creatively applied to other useful NHC-catalyzed asymmetric transformations.

[1] School of Pharmaceutical Sciences, Collaborative Innovation Center for Diagnosis and Treatment of Infectious Diseases, Key Laboratory of Bioorganic Phosphorous Chemistry and Chemical Biology (Ministry of Education), Tsinghua University Beijing, 100084 Beijing, China. [2] Division of Chemical and Life Sciences & Engineering and KAUST Catalysis Center, King Abdullah University of Science and Technology, Thuwal, 23955-6900, Saudi Arabia. Correspondence and requests for materials should be addressed to J.W. (email: wangjian2012@tsinghua.edu.cn)

Axial chirality, a key stereogenic element, is widely observed in natural products[1–3] and often determines the pharmacological properties in biologically active molecules (e.g., Maxi-K channel openers, (R)-Streptonigrin; Fig. 1)[4]. Among them, axially chiral biaryls are recognized as one of fundamental entities of chiral ligands, catalysts, and other useful reagents[5].

**Fig. 1** Representative molecules and synthetic protocols. **a** Two representative axially chiral molecules. **b** NHC-catalyzed transformations via the use of unsaturated acyl azolium intermediate. **c** Our synthetic proposal via [3+3] atroposelective annulation. NHCs react with ynals to generate chiral alkynyl acyl azolium intermediates to further react with cyclic 1,3-diones

**Table 1 Optimization of the reaction conditions[a]**

| Entry | Deviation from standard conditions[a] | Yield 3 (%)[b] | er 3 (%)[c] | Yield 4 (%)[b] | Yield 5 (%)[b] | Yield 6 (%)[b] |
|-------|---------------------------------------|----------------|-------------|----------------|----------------|----------------|
| 1 | None | 70[d] | 90:10 | <5 | <5 | <5 |
| 2 | No cat. **A** | 0 | – | 0 | 0 | 0 |
| 3 | **B** instead of **A** | <5 | – | <5 | <5 | <5 |
| 4 | **C** instead of **A** | <5 | – | <5 | <5 | 60 |
| 5 | **D** instead of **A** | 44 | −77:23 | <5 | 30 | 20 |
| 6 | No Mg(OTf)$_2$ | 60 | 90:10 | <5 | <5 | 18 |
| 7 | LiCl instead of Mg(OTf)$_2$ | 58 | 90:10 | <5 | <5 | <10 |
| 8 | In(OTf)$_3$ instead of Mg(OTf)$_2$ | 63 | 90:10 | <5 | <5 | <10 |
| 9 | Sc(OTf)$_3$ instead of Mg(OTf)$_2$ | 60 | 90:10 | <5 | <5 | <10 |
| 10 | Zn(OTf)$_2$ instead of Mg(OTf)$_2$ | 61 | 90:10 | <5 | <5 | <10 |
| 11 | CHCl$_3$ as solvent | 40 | 80:20 | <5 | 15 | 23 |
| 12 | THF as solvent | 54 | 85:15 | <5 | <10 | 19 |
| 13 | Dioxane as solvent | 20 | – | <5 | <5 | 50 |
| 14 | No $^n$Bu$_4$NOAc | 0 | – | 0 | 0 | 0 |
| 15 | DIPEA as base | 45 | 80:20 | <5 | 17 | 20 |
| 16 | Cs$_2$CO$_3$ as base | 40 | 75:25 | <5 | 20 | 22 |
| 17 | KO$^t$Bu as base | <10 | – | <5 | <5 | <5 |
| 18 | **F** instead of **E** | <5 | – | 76 | <5 | <5 |
| 19 | **G** instead of **E** | <5 | – | 70 | <5 | <5 |
| 20 | 10 mol% **A** | 69[e] | 91:9 | <5 | <5 | 13 |
| 21 | **2b** instead of **2a** | 60[f] | 75:25 | <5 | <5 | 18 |
| 22 | **2c** instead of **2a** | 60[g] | 71.5:28.5 | <5 | <5 | <5 |

[a]Standard conditions: **1a** (0.11 mmol), **2a** (0.10 mmol, R = Me), $^n$Bu$_4$NOAc (0.2 mmol), oxidant **E** (0.15 mmol), Mg(OTf)$_2$ (20 mol%), cat. **A** (15 mol%), toluene (2.0 mL), room temperature, N$_2$, 24 h
[b]Isolated yield
[c]Determined by chiral HPLC
[d]**3aa** as major product
[e]48 h
[f]**3ab** as major product
[g]**3ac** as major product

NHC:

**A:** Ar = 2,4,6-(Br)$_3$-C$_6$H$_2$
**B:** Ar = Mes
**C:** Ar = C$_6$F$_5$

**D**

Oxidant:

**E**

**F**

MnO$_2$    **G**

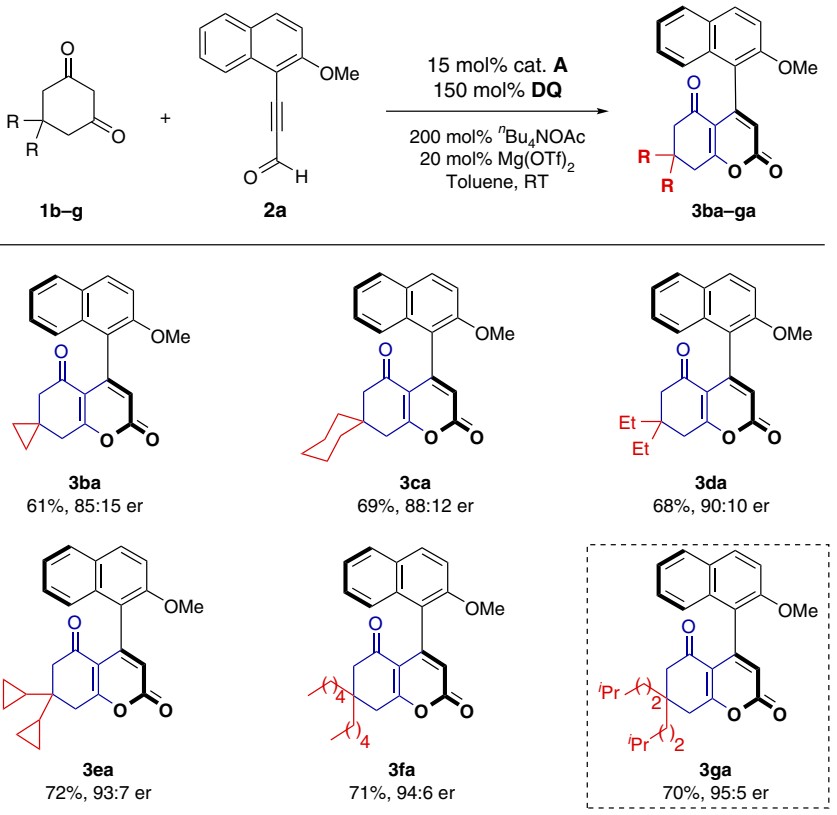

**Fig. 2** Scope of cyclic 1,3-diones. Reaction conditions: a mixture of **1b**–**g** (0.11 mmol), **2a** (0.10 mmol), $^n$Bu$_4$NOAc (0.2 mmol), oxidant **E** (0.15 mmol), Mg (OTf)$_2$ (20 mol%), and cat. **A** (15 mol%) in toluene (2.0 mL) was stirred at room temperature under N$_2$ for 24 h

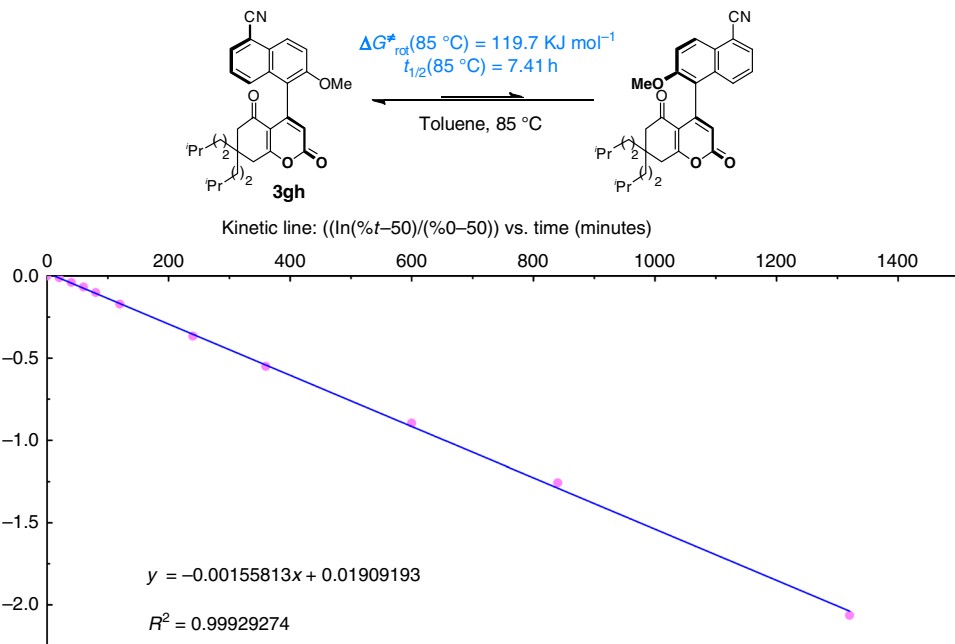

$$\Delta G^{\ddagger}_{rot}(85\ °C) = 119.7\ KJ\ mol^{-1}$$
$$t_{1/2}(85\ °C) = 7.41\ h$$

Kinetic line: $((\ln(\%t-50)/(\%0-50))$ vs. time (minutes)

$y = -0.00155813x + 0.01909193$

$R^2 = 0.99929274$

**Fig. 3** Determination of the enantiomerization barrier. Reaction conditions: 3 mg of enantio-enriched **3gh** were refluxed in 15 mL of toluene at 85 °C. Samples of 7 µL of this solution were injected on Chiralpak IC (heptane/iPrOH = 80/20, 1 mL min$^{-1}$, UV detection at 254 nm) to monitor the percentage decrease of the second eluted enantiomer over time

Over the past few decades, numerous efforts have been devoted to constructing these axially chiral biaryls, but successful examples are relatively scarce in contrast to their great potential in various applications[6-25]. In 1984, Meyers and coworkers reported the first example of central-to-axial chirality conversion in biarylic systems[26]. Later on, the direct asymmetric cross-coupling of two aryls has proven to be a feasible method[27-33]. However, the poor enantiocontrol and low coupling efficiency greatly limit their applications. More recently, an elegant route to synthesize axially chiral biaryls was demonstrated via an aromatic ring

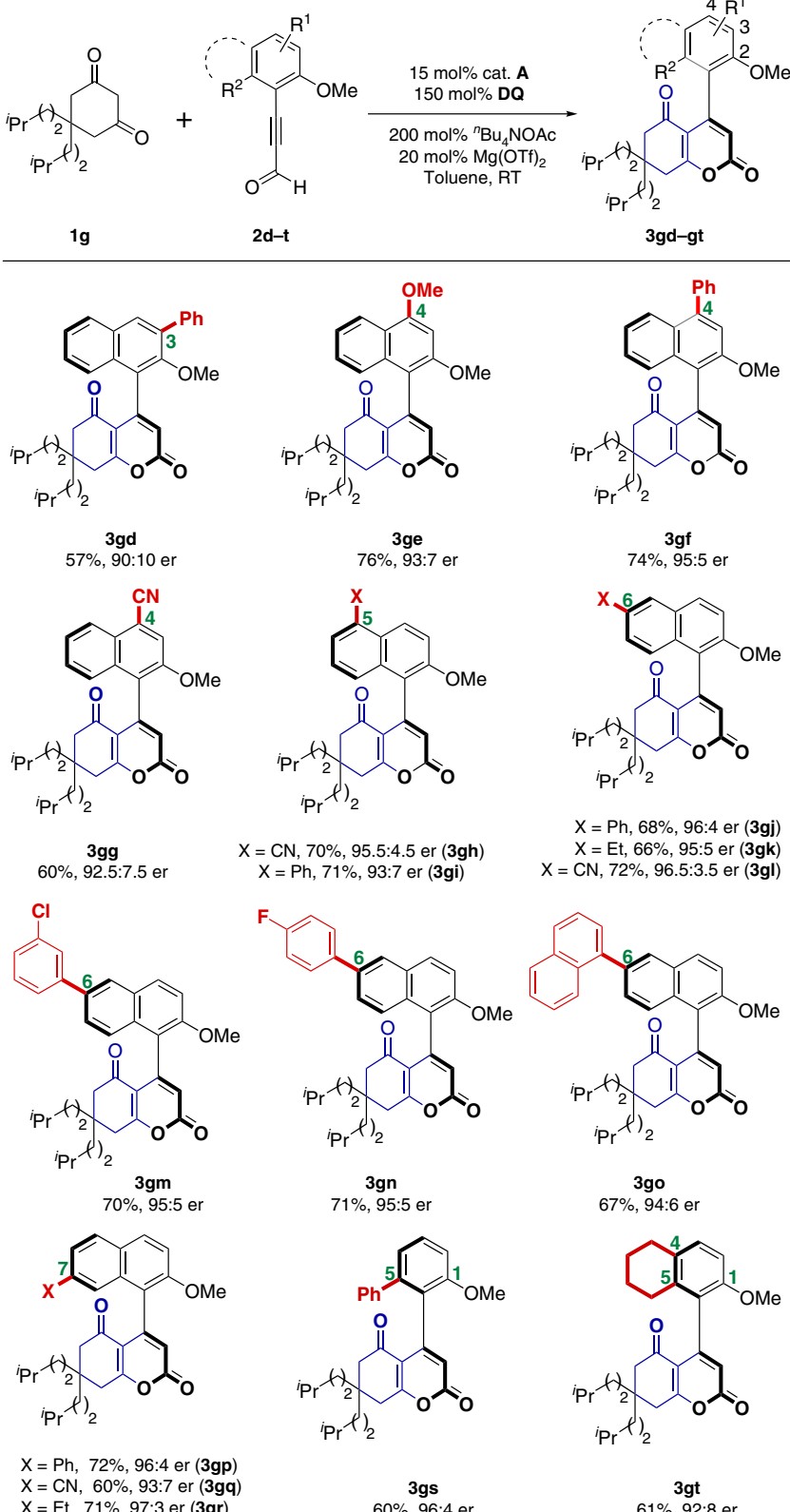

**Fig. 4** Scope of ynals. Reaction conditions: a mixture of **1g** (0.11 mmol), **2d–t** (0.10 mmol), *ⁿ*Bu₄NOAc (0.2 mmol), oxidant **E** (0.15 mmol), Mg(OTf)₂ (20 mol%), and cat. **A** (15 mol%) in toluene (2.0 mL) was stirred at room temperature under N₂ for 24 h

formation[34,35]. Despite these advances, this field is still in its infancy and efficient synthetic routes still need to be identified.

Chiral *N*-heterocyclic carbenes (NHCs) as versatile catalysts have been well studied in last few decades[36–43], but most of the reports are only focused on the assembly of central chirality.

Herein, we report a highly enantioselective NHC-catalyzed [3+3] atroposelective annulation of ynals with cyclic 1,3-diones[44], thus paving a route toward axially chiral biaryls. It is noteworthy that the NHC-bounded alkynyl acyl azoliums as active intermediates are generated from ynals in contrast to unsaturated acyl azoliums

**Fig. 5** Scope of Diels–Alder reaction. Reaction conditions: a mixture of **8** (0.1 mmol), **9** (1.0 mmol), in toluene (2.0 mL) was stirred at room temperature for 72 h

(Fig. 1) made from ynals via an internal redox reaction, which have been intensively investigated in organic reactions, such as esterification, Claisen rearrangement, cycloaddition, etc[45–55]. Our mechanistic studies have completely ruled out the route, involving the formation of unsaturated acyl azolium followed by a central-to-axial chiral conversion.

## Results

**Reaction optimization**. We began our study with the model reaction of 5,5-dimethylcyclohexane-1,3-dione (**1a**) and 3-(2-methoxynaphthalen-1-yl)propiolaldehyde (**2a**). Key results are briefly summarized in Table 1. Using $^n$Bu$_4$NOAc as the base, Mg (OTf)$_2$ as the additive[56,57], **E** as the oxidant, and toluene as the solvent, a number of chiral NHC catalysts **A–D**[58–62] were initially screened. No desired product was detected in the presence of widely used NHC catalysts **B** and **C**. Pleasingly, chiral triazolium NHC precatalyst with $N$-2,4,6-(Br)$_3$C$_6$H$_2$ substituent (Table 1, **D**) provided axially chiral **3aa** with a moderate er, but albeit in a low yield (Table 1, entry 5). Along with the formation of **3aa**, byproducts of **4aa**, **5aa**, and **6aa**, which resulted from different unexpected intermediates and reaction pathways, were produced simultaneously. Given the significance of reaction conditions to the success of a focused catalytic transformation, we carried out a comprehensive optimization of reaction parameters. As outlined in Table 1, addition of **1a** and **2a** to a mixture of catalyst **A** (15 mol%), oxidant **E** (1.5 equiv.), and $^n$Bu$_4$NOAc (2.0 equiv.) with Mg(OTf)$_2$ (20 mol%), provided **3aa** in 70% yield and 91:9 er (Table 1, entry 1).

**Substrate scope**. With the most efficient catalytic conditions in hand, we next examined the substrate scope (Fig. 2). The R substituent of cyclic 1,3-dione **1** was investigated firstly. Substrates equipped with cyclic substituents (e.g., four- and six-membered rings) on cyclic 1,3-dione scaffold gave the corresponding products **3ba** and **3ca** in good yields but only with moderate er. In addition, reactions for cyclic 1,3-dione substrates bearing alkyl chains in different length proceeded smoothly under standard reaction conditions (**3da–fa**). While substrate cyclic 1,3-dione (**2g**) bearing a long alkyl chain was used, a good yield and high er value were achieved (Fig. 2, **3ga**, 70% yield and 95:5 er).

To address the stability of the products, we conducted several experiments and the related results verified that the rotation barrier of the chiral axis was high enough to prevent the racemization of product **3gh** during the reaction or its purification: with $\Delta G^{\ddagger}_{rot} = 119.7$ KJ mol$^{-1}$ at 85 °C, the half-life of rotation is 7.41 h at 85 °C (Fig. 3; for details, see Supplementary Discussion).

Further investigation on the scope of ynals was conducted (Fig. 4). The steric and electronic effects on the aromatic ring of ynals were evaluated by the variation of substituent patterns. When examined substrates bear electron-withdrawing or electron-donating groups at 3-, 4-, 6-, 7-, or 8-substituted positions on naphthalene rings, moderate to good yields and high er values were regularly obtained (**3gd–gr**). When a substituted phenyl ring replaced the naphthalene ring in ynals, high er could still be achieved (**3gs** and **3gt**). The absolute configuration of **3au** was determined to be (*S*) by X-ray crystallography, and other products were assigned by analogy.

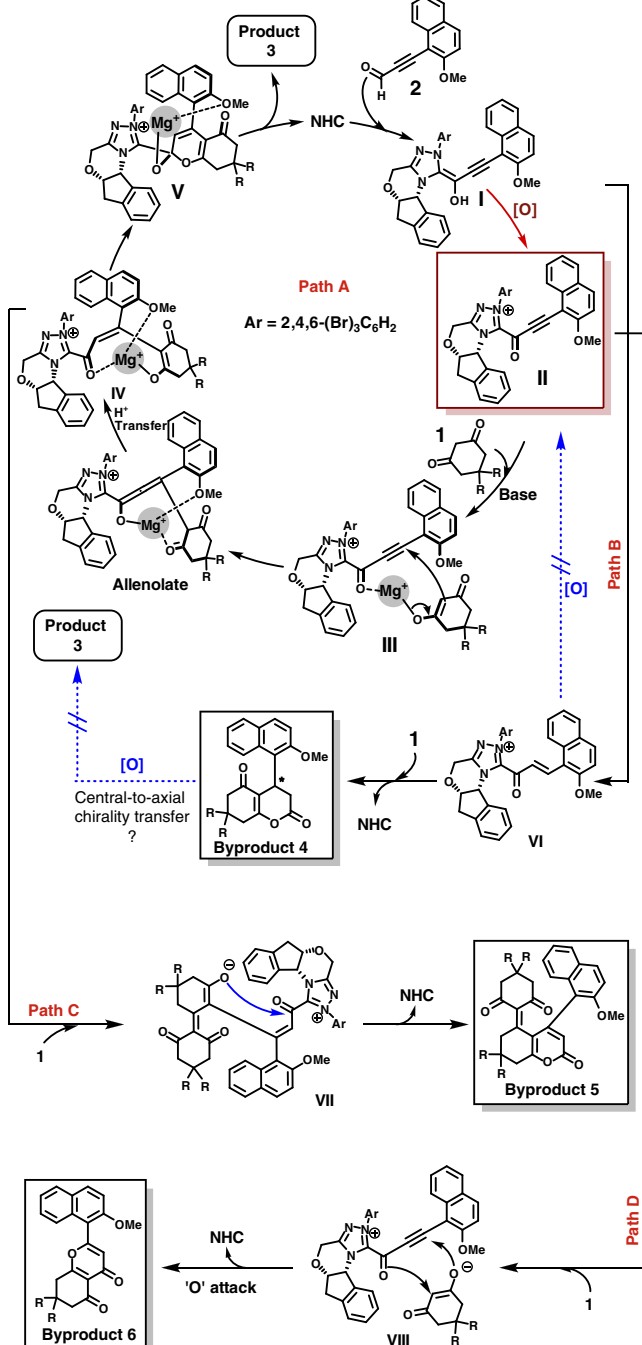

**Fig. 6** Postulated mechanistic pathways. Path A shows the formation of product **3**. Path B suggests the formation of byproduct **4**. Path C is a probable way to generate byproduct **5**. Path D indicates a plausible route to explain the formation of byproduct **6**

To demonstrate the utility of above synthesized products, we successfully converted **8** into commonly used axially biaryls **10**. As shown in Fig. 5, Diels–Alder reaction of **8** and **9** afforded the corresponding axially chiral naphthyl–phenyl products **10** in acceptable yields and no racemization was observed.

**Mechanistic studies.** The origins of chemo- and stereo-selectivity of this reaction are rationalized by the postulated mechanism illustrated in Fig. 6 (Path A). The addition of NHC catalyst to ynal **2** yields an NHC-bounded Breslow intermediate **I**[63,64].

Breslow intermediate **I** then undergoes oxidation to generate the firstly proposed intermediate, alkynyl acyl azolium **II**, which subsequently reacts with cyclic 1,3-dione **1** to form intermediate **III**. **III** undergoes Michael addition to the alkynyl azolium moeity to form the allenolate intermediate and after subsequent proton transfer from the 1,3-dione to the allene, intermediate **IV** is reached. Next O–C bond is formed to create **V** and the NHC can be released and finally generated product **3**. As the generation of NHC-bounded unsaturated acyl azolium intermediates from ynals has been reported by Zeitler[45], Lupton[46,47], Bode[48–51], Scheidt[52], and others[53–55], an alternative pathway may involve the direct annulation of NHC-bounded unsaturated acyl azolium intermediate **VI** with cyclic 1,3-dione **1** leading to byproduct **4**. However, as highlighted in Fig. 7 (Eq. (1)), the oxidative dehydrogenation of **4aa** to **3aa** does not proceed in the presence of oxidant alone or under standard reaction conditions. As such, **3aa** cannot be generated from the α,β-unsaturated acyl azolium intermediate.

During the process of optimization, byproduct **5** was found clearly and confirmed by NMR spectra, presumably generated through the Knoevenagel condensation of **3** with 1.0 equivalent of **1**. To examine this hypothesis, a controlled experiment was carried out (Fig. 7, Eq. (3)). Surprisingly, the er value of **7** is not consistent with the er value of **3gs** (59:41 er vs. 96:4 er) and this observation indicates that an alternative pathway may be operating (Fig. 6, Path C). Building upon intermediate **IV**, we suggest that the Knoevenagel condensation process generates intermediate **VI** which subsequently leads to **5** via annulation. Moreover, there is an interesting observation found during the optimization of reaction conditions with Lewis acids (Table 1, entry 6). When $Mg(OTf)_2$ is omitted from the reaction condition, the yield of byproduct **6** increases to 18%, which can be explained by the fact that **1** can now do a direct 'O' attack to the alkynyl on intermediate **II**, because the $Mg^{2+}$ ion is not there to coordinate **1** and **II**. Therefore, $Mg^{2+}$ plays a critical role as it reduces the ketoenolate's 'O' attack (transition state **VIII**, Path D) and promotes the 'C' attack (intermediate **III**, Path A, Fig. 6).

Preliminary computational studies were conducted on steps **III** to **V** in Path A assuming an acetate ligand on the magnesium ion to provide insights into the observed enantioselectivity. It was found that the energies of all transition states from **III** to the allenalate are higher than those of the rest of processes and we thus hypothesize that the enentioselectivity is determined in this intramolecular C–C bond forming reaction. Interestingly, in contrast to other studies on the α,β-unsaturated acyl azolium analogs, this step creats two components of axial chirality, namely the allenolate and the 2-methoxynaphthalen-1-yl moiety, in addition to one chiral center of the 1,3-dione. The twisted alkynyl acyl azolium plane allows the ketoenolate group to stay away from the indane ring (Fig. 8), whose role is to discriminate the strain energy during the formation of the allenolate center rather than the intuitive effect to block the approach of the nucleophile.

## Discussion

In summary, we have successfully developed an NHC-catalyzed atroposelective annulation of cyclic 1,3-diones with ynals, providing chiral α-pyrone-aryls in moderate to good yields with high enantioselectivities. This protocol features good functional group tolerance, and allows the rapid assembly of axially chiral molecules from simple and readily available starting materials under mild conditions. Our computational investigation suggests that the enantioselectivity is determined during the Michael addition of the ketoenolate to the alkynyl azolium moiety. Further investigations on axially chiral compounds as hits in medicinal chemistry or as chiral ligands or catalysts in asymmetric

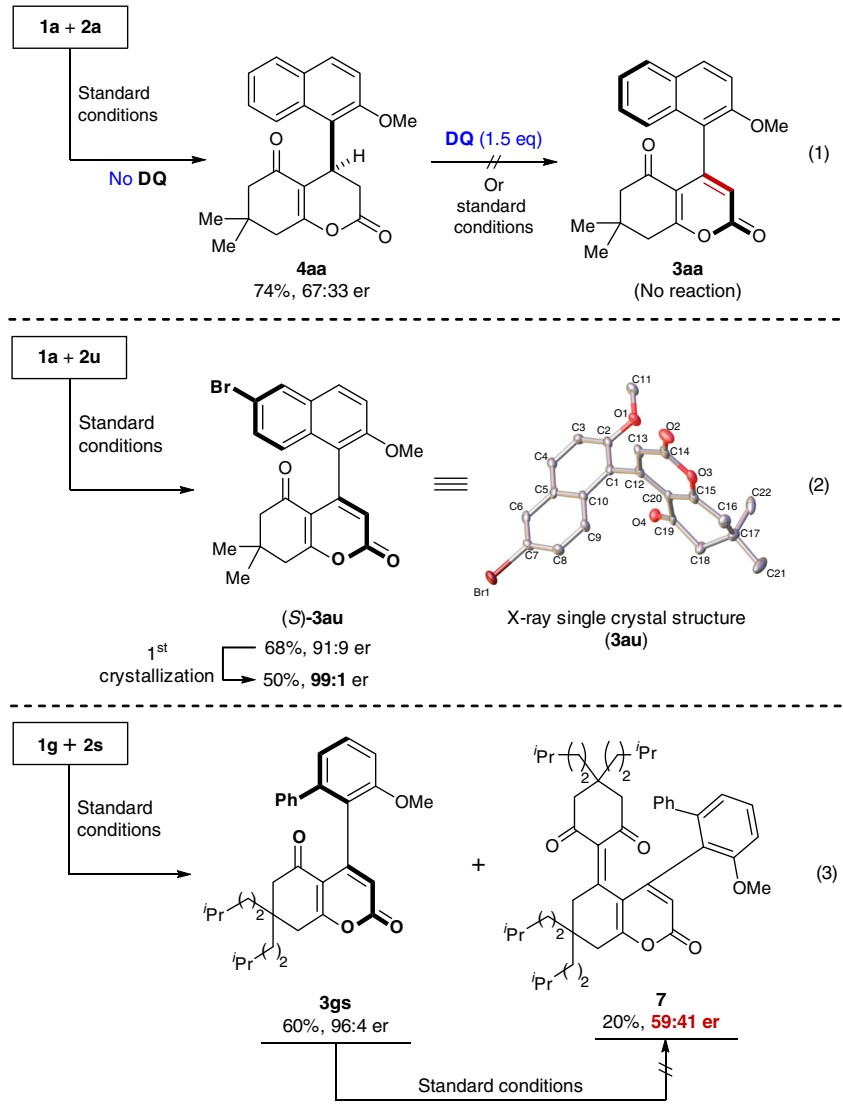

**Fig. 7** Control experiments. (1) **4aa** failed to undergo oxidation to form **3aa** in the presence of DQ. (2) The absolute configuration of 3au was determined to be (S) by X-ray crystallography. **(3)** Under standard conditions, the reaction of **1g** with **2s** yielded **3gs** and **7**. However, we found that **7** was not directly generated from **3gs** under currrent conditions

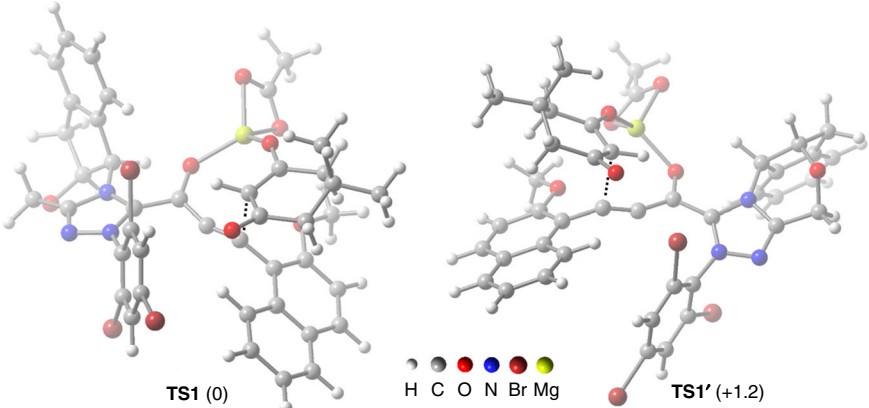

**Fig. 8** Comparison of transition states. Relative free energy (kcal mol⁻¹) of **TS1** and **TS1'** are displayed in the brackets

synthesis, as well as a detailed mechanistic study, are currently underway in our laboratories.

## Methods

**Synthesis of 3**. In a glovebox, a flame-dried Schlenk reaction tube equipped with a magnetic stir bar, NHC precatalyst **A** (9.2 mg, 0.015 mmol), $^nBu_4NOAc$ (60.2 mg, 0.20 mmol), oxidant DQ (62.0 mg), cyclic 1,3-dione **1** (0.11 mmol), ynal **2** (0.10 mmol), and freshly distilled toluene (2.0 mL) were added. The reaction mixture was stirred at room temperature for 24 h. The mixture was then filtered through a pad of Celite washed with DCM. After the solvent was evaporated, the residue was purified by flash column chromatography to afford the desired product **3**.

**Computational details**. All structures and energies were computed using the Gaussian 09 program package version D.01[65]. The B3LYP functional together with the 6-31g(d,p) basis set was used. All structures were optimized to a minimum confirmed by frequency calculations and all transition state structures were confirmed by identifying one imaginary frequency and intrinsic reaction coordinate (IRC) analysis.

**Data availability**. For $^1H$, $^{13}C$ NMR, and high-performance liquid chromatography spectra of the compounds in this manuscript, see Supplementary Figs. 1–167. For the details of the synthetic procedures, see Supplementary Methods. The supplementary crystallographic data for this paper could be obtained free of charge from The Cambridge Crystallographic Data Centre (**3au**: CCDC 1501039) via https://www.ccdc.cam.ac.uk/

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

## Acknowledgements

Generous financial supports for this work were provided by: the National Natural Science Foundation of China (21672121), the "Thousand Plan" Youth program of China, the Tsinghua University, the Bayer Investigator fellow, the fellowship of Tsinghua-Peking centre for life sciences (CLS), and the China Postdoctoral Science Foundation (2015M570072) to J.W., and KAUST to K.-W.H.

## Author contributions

C.Z. conducted the main experiments; F.L. and D.G. prepared several starting materials, including substrates and NHC catalysts. K.M. and K.-W.H. conducted the computational studies. J.W. conceptualized and directed the project, and drafted the manuscript with the assistance from co-authors. All authors contributed to discussions.

## Additional information

**Competing interests:** The authors declare no competing financial interests.

