## [Peer Review File · Nature Communications]

Reviewers' comments:

Reviewer #1 (Remarks to the Author):

The manuscript by Wang reports the annulation of ynals acyl azoliums with 1,3-dicarbonyls, specifically 1,3-cyclohexadione and closely related derivatives. The reaction provides axially chiral pyranones in good to excellent levels of enantioselectivity, with an array of examples reported. A derivatisation and mechanistic studies are reported. The most convincing of the later relates to an inability of the conditions to convert the dihydropyranone to the product of this reaction.

The manuscript is relatively well presented, although some reordering of text and images is needed as it was always clear how the text figures linked.

The novelty resides in the use of the ynals acyl azolium. To my reading of the literature this is the first example of its use in NHC catalysis. This is not the first example of a NHC catalysed formation of axially chiral materials (Lupton, ref 44) have reported this previously, although this is a point to axial example, but it is likely the first atropselective reaction, so this claim of novelty is also likely warranted.

In general I think that the reaction reported has been studied at a depth that makes it appropriate for publication and the novelty is sufficient for this journal. Thus, I recommend publication after a few issues are addressed:

1) the scope in regard to the dicarbonyl is remarkably narrow. Do other cyclic 1,3-dicarbonyls work. β -keto ester etc. Expanded scope and/or comments regarding the limitations required.

The mechanism seems reasonable. An additional experiment with 2 (or 3) equiv of DQ with cinamaldehyde is required. These conditions should favour the formation of the point chiral intermediate dihydropyranone and then its potential conversion to the axially chiral product can be ascertained.

Reviewer #2 (Remarks to the Author):

In this manuscript, Wang and co-workers reported a chiral NHC-catalyzed asymmetric annulation of cyclic 1, 3-diones with ynals, generating diverse axially chiral α -pyrone-aryls in moderate to good yields and enantioselectivities. In addition, the products are also easily transformed into axially chiral biaryl derivatives. The chemoselectivity was well controlled by screening the NHC pre-catalysts, bases, Lewis acids, oxidants and solvents. The chiral alkynyl acyl azoliums was described as the key active intermediates, which was used to synthesize axially chiral compounds with moderate to good results for the first time. It was a good development in the field of NHC catalysis. The nice work seems to have been well conducted with sufficient details, and can be of utility to researchers interested in the organic chemistry and medicinal chemistry. Thus I recommend the publication of this manuscript in Nature Communications after the following issues addressed:

1. Can derivatives of cyclohexane-1, 3-dione without substituents at 5-position or acyclic diones be compatible substrates in the reactions?

2. The transformation obtained products into biaryls via Diels-Alder reaction of ynone-aryls and alkyne was very interesting and applicable. More examples should be investigated to demonstrate the versatility.
3. The absolute configuration(R or S) of 3au should be given.
4. The number of compounds should be in a uniform format. The ee of compound 10 should not be italic and should display in the same line with yield.
5. Some errors in the article: (Table 1, entry 4) should be (Table 1, entry 5); (see Scheme 4, proposed mechanisms) should be (see Scheme 3, proposed mechanisms); ref: 45-45 (scheme 1a) should be ref: 45-55; 76% yield (Table 1, entry 4 and SI19, entry 18) should be 74% yield (from S36); 63% yield (Table 1, entry 21) may be 60% yield (SI19, entry 30); 22% yield (Table 1, entry 22) may be < 5% yield (SI19, entry 30); 94:8 er (Table 2, 3fa) should be 94:6 er; at 80 oC in article (below Scheme 2) should be 85 oC; substituted positions on naphthalene rings in article and table 3 are incorrect; intermediate 4 should be byproduct 4; (18% ee vs 90% ee) should be (18% ee vs 92% ee); 3au 66% yield should be 68%.
6. Chemical structure of 8 is wrong (SI 38) and 3gs(S35) should be 3gq.
7. The NMR analytical errors: 2j; 2l; 2n; 2u; 2p-2t; 3aa; 3ac. Part of the resulting products need to re-purify for better NMR spectra(3ab; 3ba; 3ca; 3gd et.al).
8. Ee value errors (SI): 3gg; 3aa; 3ac; 3ea; 3ge; 3gi. (e.g. 3gi er 96:4 should be 93:7).
9. The paper itself requires very careful proof reading and copy-editing to ensure the language is appropriate and correct.
10. A related paper should be cited: Nat. Commun. 2017, 8, 15238. DOI: 10.1038/ncomms15238.

Reviewer #3 (Remarks to the Author):

The manuscript from Wang et al describes an enantioselective annulation of alkynyl aldehydes and 1,3-diketones to generate axially chiral molecules under the auspices of a chiral N-heterocyclic carbene. These are interesting and valuable products and this represents an interesting strategy. The reaction itself has reasonable scope, with 18 examples in total, indicating a fair tolerance for substitution around the naphthol ring (and some relatively trivial tolerance upon the 1,3-diketone). There is no example that has a substituent adjacent to the OMe group on the naphthol, which is a surprise, as it could be expected that this would be the most desirable position. Is this because this substrate is ineffective in the reaction or because it cannot or has not been made?

The catalyst loading is high (at 15%) – is there any reason for this? (entry 20 in table 1 indicated the same yield and e.r. as 10% loading).

The most interesting part of the paper is the proposed mechanism involving an alkynyl acyl azolium intermediate. The authors suggest that this is an intermediate through a series of control experiments described in equations 2,3 and 4. These are imperfect experiments as they do not necessarily reproduce the 'standard' conditions (as these conditions do not contain the product 3aa or the phenolic byproduct of the oxidation reaction). Application of DFT or other suitable calculation methods could also help to elucidate the mechanism; this would add significantly to the impact of the paper.

The authors determine the barrier to rotation at 80 °C (to be 119.7 kJmol⁻¹) and then extrapolate this to a half-life at 25 °C without considering entropy. This is inappropriate and the half-life should be stated at the temperature at which it was determined.

My overall view is that this is a good paper and with appropriate attention could be suitable for publication in nature communications.

Trivial comments:

The paper needs to be carefully proof read for appropriate use of grammar and language, as there are many mistakes throughout.

The authors switch between e.e. and e.r. throughout the paper. Either is fine in my view but the authors should stick to one.

Based on reviewers' comments, we have fully revised our manuscript accordingly and all the changes in the main manuscript are highlighted *in color*. The detailed point-to-point responses to the reviewer's comments and our changes are listed below.

Response to reviewer 1:

1. The manuscript by Wang reports the annulation of ynal acyl azoliums with 1,3-dicarbonyls, specifically 1,3-cyclohexadione and closely related derivatives. The reaction provides axially chiral pyranones in good to excellent levels of enantioselectivity, with an array of examples reported. A derivatisation and mechanistic studies are reported. The most convincing of the later relates to an inability of the conditions to convert the dihydropyranone to the product of this reaction.

Answer:

We are very pleased to thank this reviewer to give this very positive response.

2. The manuscript is relatively well presented, although some reordering of text and images is needed as it was always clear how the text figures linked.

Answer:

We followed this suggestion to fully revise our manuscript including reordering of some text and images. Please check our revised manuscript, especially the yellow color.

3. The novelty resides in the use of the ynal acyl azolium. To my reading of the literature this is the first example of its use in NHC catalysis. This is not the first example of a NHC catalysed formation of axially chiral materials (Lupton, ref 44) have reported this previously, although this is a point to axial example, but it is likely the first atropselective reaction, so this claim of novelty is also likely warranted.

Answer:

We followed this suggestion and revised the abstract accordingly. For example, "*Control experiments indicated that alkynyl acyl azoliums, as novel active intermediates, are employed to atropselectively assemble axially chiral biaryls and such a methodology may be creatively applied to other useful NHC-catalyzed asymmetric transformations*".

4. The scope in regard to the dicarbonyl is remarkably narrow. Do other cyclic 1,3-dicarbonyls work. β -keto ester etc. Expanded scope and/or comments regarding the limitations are required.

Answer:

We have examined other 1,3-dicarbonyls, such as cyclohexane-1,3-dione (H at 5-position), pentane-2,4-dione, β -keto ester (e.g. ethyl 3-oxobutanoate, ethyl 3-oxo-3-phenylpropanoate), and 4-hydroxycoumarin. However, very complicated mixture was obtained in above experiments. And also, the mixture was difficult to identify and isolate. The main reason led to this phenomenon may be the use of a

large amount of oxidants. Similar results have also been reported by Rodriguez and coworkers recently. For details, please see: *Angew. Chem. Int. Ed.* **2016**, *55*, 1401-1405.

5. The mechanism seems reasonable. An additional experiment with 2 (or 3) equiv of DQ with cinamaldehyde is required. These conditions should favour the formation of the point chiral intermediate dihydropyranone and then its potential conversion to the axially chiral product can be ascertained.

Answer:

Actually, a similar experiment was already described as eqn (1) in our manuscript (Page 4). We have carried out the experiment by use of 3 equiv. of oxidant DQ. As a result, compound **4aa** was obtained in 80% yield. And no axially chiral product **3aa** was observed based on TLC, crude ^1H NMR, and LC-MS detection. In addition, we also conducted the requested experiment with cinamaldehyde suggested by this reviewer. As you can see below, only product **11** was observed. The related information has been placed in SI (See Page 26 in SI).

Additional experiment:

Response to reviewer 2:

1. In this manuscript, Wang and co-workers reported a chiral NHC-catalyzed asymmetric annulation of cyclic 1, 3-diones with ynals, generating diverse axially chiral α -pyrone-aryls in moderate to good yields and enantioselectivities. In addition, the products are also easily transformed into axially chiral biaryl derivatives. The chemoselectivity was well controlled by screening the NHC pre-catalysts, bases, Lewis acids, oxidants and solvents. The chiral alkynyl acyl azoliums was described as the key active intermediates, which was used to synthesize axially chiral compounds with moderate to good results for the first time. It was a good development in the field of NHC catalysis. The nice work seems to have been well conducted with sufficient details, and can be of utility to researchers interested in the organic chemistry and medicinal chemistry. Thus I recommend the publication of this manuscript in *Nature Communications* after the following issues addressed:

Answer:

Thanks this reviewer for this very positive commence, wo have followed all suggestions to revise our manuscript.

2. Can derivatives of cyclohexane-1, 3-dione without substituents at 5-position or acyclic diones be compatible substrates in the reactions?

Answer:

We have examined other 1,3-dicarbonyls, including cyclohexane-1,3-dione (H at 5-position), pentane-2,4-dione, β -keto ester (e.g. ethyl 3-oxobutanoate, ethyl 3-oxo-3-phenylpropanoate), and 4-hydroxycoumarin. However, very complicated mixture was obtained in above experiments. And also, the mixture was difficult to identify and isolate. The main reason led to this phenomenon may be the use of a large amount of oxidants. Similar results have also been reported by Rodriguez and coworkers recently. For details, please see: *Angew. Chem. Int. Ed.* **2016**, *55*, 1401-1405.

3. The transformation obtained products into biaryls via Diels-Alder reaction of ynone-aryls and alkyne was very interesting and applicable. More examples should be investigated to demonstrate the versatility.

Answer:

As requested by reviewer, we have expanded the scope of Diels-Alder reaction (See below). Please also check this Table 3 in our revised manuscript.

Table 3: Scope of Diels-Alder reaction

4. The absolute configuration (R or S) of 3au should be given.

Answer:

The absolute configuration of 3au is (S). We have added this information in our

revised manuscript (Page 2, “The absolute configuration of **3au** was determined to be (*S*) by X-ray crystallography (eqn (2))” and Page 4, equation 3; and SI.

5. The number of compounds should be in a uniform format. The ee of compound 10 should not be italic and should display in the same line with yield.

Answer:

We have changed the format in our revised manuscript accordingly. Note: we used “er” to replace “ee” in our revised manuscript based on reviewer’s suggestion.

6. Some errors in the article: (Table 1, entry 4) should be (Table 1, entry 5); (see Scheme 4, proposed mechanisms) should be (see Scheme 3, proposed mechanisms); ref: 45-45 (scheme 1a) should be ref: 45-55; 76% yield (Table 1, entry 4 and SI19, entry 18) should be 74% yield (from S36); 63% yield (Table 1, entry 21) may be 60% yield (SI19, entry 30); 22% yield (Table 1, entry 22) may be < 5% yield (SI19, entry 30); 94:8 er (Table 2, 3fa) should be 94:6 er; at 80 oC in article (below Scheme 2) should be 85 oC; substituted positions on naphthalene rings in article and table 3 are incorrect; intermediate 4 should be byproduct 4; (18% ee vs 90% ee) should be (18% ee vs 92% ee); 3au 66% yield should be 68%.

Answer:

All these errors have been corrected in the revised manuscript or SI.

7. Chemical structure of 8 is wrong (SI 38) and 3gs(S35) should be 3gq.

Answer:

Chemical structure of 8 has been corrected and 3gs has been changed to 3gq in the revised SI.

8. The NMR analytical errors: 2j; 2l; 2n; 2u; 2p-2t; 3aa; 3ac. Part of the resulting products need to re-purify for better NMR spectra (3ab; 3ba; 3ca; 3gd et.al).

Answer:

The NMR of 2j; 2l; 2n; 2u; 2p-2t; 3aa; 3ac have been reanalyzed and 3ab; 3ba; 3ca; 3gd have been re-purified. Updated NMR of 3ab; 3ba; 3ca; 3gd has been added to SI.

9. Ee value errors (SI): 3gg; 3aa; 3ac; 3ea; 3ge; 3gi. (e.g. 3gi er 96:4 should be 93:7).

Answer:

The ee value errors of 3gg; 3aa; 3ac; 3ea; 3ge; 3gi have been corrected in the revised SI.

10. The paper itself requires very careful proof reading and copy-editing to ensure the language is appropriate and correct.

Answer:

Thank you very much for your advice, we have checked our manuscript carefully. The typos and use of incorrect words have been corrected in our revised

manuscript.

11. A related paper should be cited: Nat. Commun. 2017, 8, 15238. DOI: 10.1038/ncomms15238.

Answer:

We have added this reference in the revised manuscript. Please see new reference 25.

Response to reviewer 3:

1. The manuscript from Wang et al describes an enantioselective annulation of alkynyl aldehydes and 1,3-diketones to generate axially chiral molecules under the auspices of a chiral N-heterocyclic carbene. These are interesting and valuable products and this represents an interesting strategy. The reaction itself has reasonable scope, with 18 examples in total, indicating a fair tolerance for substitution around the naphthol ring (and some relatively trivial tolerance upon the 1,3-diketone).

Answer:

Thanks this reviewer for this very positive comment.

2. There is no example that has a substituent adjacent to the OMe group on the naphthol, which is a surprise, as it could be expected that this would be the most desirable position. Is this because this substrate is ineffective in the reaction or because it cannot or has not been made?

Answer:

This reviewer asked a very impressive and critical question. Actually, we have synthesized the following three ynals (see below figure, using Me or Ph to replace OMe). Unfortunately, these reactions are very sluggish and we obtained a complex using our standard conditions which was difficult to determine which one is which. Meanwhile, we also did a DFT calculation (see new manuscript), and the result told us the OMe group is important for coordination with Mg²⁺. Thus, the OMe group is an important and necessary substituent in the assembly of axially chiral biaryls.

Another example is indicated below. We tried to optimize the reaction conditions to achieve the desired product. After the optimization of base, solvent and NHC

catalyst, the propargyl ester was obtained as the major product. This result also proved the existence of OMe on the flip side is necessary.

Meanwhile, the DFT calculation also suggests that the OMe group will provide a coordination with Lewis acid $Mg(OTf)_2$ (Please Scheme 3 in Manuscript). This important coordination would assist the annulation to give the final biaryl products. When $Mg(OTf)_2$ is omitted from the reaction condition, the yield of byproduct **6** increases to 18%, which can be explained by the fact that **1** can now do a direct “O” attack to the alkynyl on intermediate **II**, because the Mg^{2+} ion is not there to coordinate **1** and **II**. Therefore, Mg^{2+} plays a critical role as it reduces the ketoenolate’s “O” attack (transition state **VIII**, Path D) and promotes the “C” attack (intermediate **III**, Path A, Scheme 3).

Scheme 3: Postulated mechanistic pathways

3. The catalyst loading is high (at 15%), is there any reason for this? (entry 20 in table 1 indicated the same yield and e.r. as 10% loading).

Answer:

A 10 mol % catalyst loading can lead to ab.13% increase in byproduct **6**, which will also cause a difficult purification in late-stage. That is reason why we choose 15 mol % catalyst loading.

4. The most interesting part of the paper is the proposed mechanism involving an alkynyl acyl azolium intermediate. The authors suggest that this is an intermediate through a series of control experiments described in equations 2,3 and 4. These are imperfect experiments as they do not necessarily reproduce the ‘standard’ conditions (as these conditions do not contain the product 3aa or the phenolic byproduct of the oxidation reaction). Application of DFT or other suitable calculation methods could also help to elucidate the mechanism; this would add significantly to the impact of the paper.

Answer:

Following the reviewer’s suggestion, we did computational study. The relative information has been placed in Page 4 “Preliminary computational studies were conducted on steps **III** to **V** in Path A assuming an acetate ligand on the magnesium ion to provide insights to rationalize the observed enantioselectivity. It was found that the energies of all transition states from **III** to the allenolate are higher than those of the rest of processes and we thus hypothesize that the enantioselectivity is determined in this intramolecular C-C bond forming reaction. Interestingly, in contrast to other studies on the α,β -unsaturated acyl azolium analogs, this step creates two components of axial chirality, namely the allenolate and the 2-methoxynaphthalen-1-yl moiety, in addition to one chiral center. The preferred attack was computed to occur on the side of the tribromophenyl group. The twisted alkynyl acyl azolium plane allows the ketoenolate group to stay away from the indane ring (Fig. 1), whose role is to discriminate the strain energy during the formation of the allenolate center rather than the intuitive effect to block the approach of the nucleophile.”

Figure 1. Comparison of two plausible transition states (relative free energy in kcal/mol).

5. The authors determine the barrier to rotation at 80°C (to be 119.7 kJmol⁻¹) and then extrapolate this to a half-life at 25°C without considering entropy. This is inappropriate and the half-life should be stated at the temperature at which it was determined.

Answer:

We agreed with the reviewer's opinion. We extrapolated the half-life at 85°C (see Page 2, Scheme 2). It has also been corrected in the revised manuscript and SI.

6. The paper needs to be carefully proof read for appropriate use of grammar and language, as there are many mistakes throughout.

Answer:

We have tried our best to check our manuscript carefully. The typos and use of incorrect words have been corrected in our revised manuscript. Please see our revised manuscript.

7. The authors switch between e.e. and e.r. throughout the paper. Either is fine in my view but the authors should stick to one.

Answer:

We have checked our manuscript carefully and all ee have been changed to er in our revised manuscript.

REVIEWERS' COMMENTS:

Reviewer #2 (Remarks to the Author):

The revised manuscript has taken good care of my revision suggestions with high quality. The authors have also done careful revisions according to the revisions suggestions by the other referees. For those suggested the experiments, even in some cases they are not quite straightforward, the authors have made solid attempts and enormous efforts in improving the results. Overall, the revised manuscripts has been significantly improved and should be accepted.

Reviewer #3 (Remarks to the Author):

The authors have clearly listened to the concerns and suggestions of the reviewers and have made careful and significant corrections. In my view this has improved the paper and i am now content that it is suitable for Nature Communications.